# Testing Adaptive Therapy Protocols Using Gemcitabine and Capecitabine in a Preclinical Model of Endocrine-Resistant Breast Cancer

**DOI:** 10.3390/cancers16020257

**Published:** 2024-01-06

**Authors:** Sareh Seyedi, Ruthanne Teo, Luke Foster, Daniel Saha, Lida Mina, Donald Northfelt, Karen S. Anderson, Darryl Shibata, Robert Gatenby, Luis H. Cisneros, Brigid Troan, Alexander R. A. Anderson, Carlo C. Maley

**Affiliations:** 1Arizona Cancer Evolution Center, Arizona State University, Tempe, AZ 85287, USA; 2School of Life Sciences, Arizona State University, Tempe, AZ 85287, USA; 3Biodesign Center for Biocomputing, Security and Society, Arizona State University, Tempe, AZ 85287, USA; 4Division of Biology, Kansas State University, Manhattan, KS 66506, USA; 5Division of Hematology and Oncology, Mayo Clinic Arizona, Phoenix, AZ 85054, USA; 6Center for Personalized Diagnostics, Biodesign Institute, Arizona State University, Tempe, AZ 85287, USA; 7Department of Pathology, University of Southern California Keck School of Medicine, Los Angeles, CA 90033, USA; dshibata@usc.edu; 8Department of Integrated Mathematical Oncology, Moffitt Cancer Center, Tampa, FL 33629, USAalexander.anderson@moffitt.org (A.R.A.A.); 9Department of Population Health and Pathobiology, North Carolina State University College of Veterinary Medicine, Raleigh, NC 27606, USA; 10Center for Evolution and Medicine, Arizona State University, Tempe, AZ 85287, USA

**Keywords:** adaptive therapy, endocrine-resistant MCF7 breast cancer, high-dose therapy, drug-resistant and drug-sensitive subclones

## Abstract

**Simple Summary:**

The biggest obstacle to curing cancer is the fact that cancers often harbor mutant cells, called resistant cells, that are unaffected by cancer drugs. We tested a strategy for maintaining control over resistant cells called adaptive therapy. We tested this strategy on mice that had human breast cancer. In adaptive therapy, we aim to keep alive many cancer cells that are sensitive to cancer drugs and use them to compete with the resistant cells. We can prevent the sensitive cells from growing out of control using occasional low doses of a cancer drug. Competition with the sensitive cells prevents the resistant cells from growing out of control, resulting in long-term control of the cancer. Thus, we turn cancer into a chronic, nonlethal disease. Our experiment showed the effectiveness of this approach and how we might make it even better by switching between two drugs.

**Abstract:**

Adaptive therapy, an ecologically inspired approach to cancer treatment, aims to overcome resistance and reduce toxicity by leveraging competitive interactions between drug-sensitive and drug-resistant subclones, prioritizing patient survival and quality of life instead of killing the maximum number of cancer cells. In preparation for a clinical trial, we used endocrine-resistant MCF7 breast cancer to stimulate second-line therapy and tested adaptive therapy using capecitabine, gemcitabine, or their combination in a mouse xenograft model. Dose modulation adaptive therapy with capecitabine alone increased survival time relative to MTD but not statistically significantly (HR = 0.22, 95% CI = 0.043–1.1, *p* = 0.065). However, when we alternated the drugs in both dose modulation (HR = 0.11, 95% CI = 0.024–0.55, *p* = 0.007) and intermittent adaptive therapies, the survival time was significantly increased compared to high-dose combination therapy (HR = 0.07, 95% CI = 0.013–0.42, *p* = 0.003). Overall, the survival time increased with reduced dose for both single drugs (*p* < 0.01) and combined drugs (*p* < 0.001), resulting in tumors with fewer proliferation cells (*p* = 0.0026) and more apoptotic cells (*p* = 0.045) compared to high-dose therapy. Adaptive therapy favors slower-growing tumors and shows promise in two-drug alternating regimens instead of being combined.

## 1. Introduction

Therapeutic resistance and therapeutic toxicity are two of the biggest challenges in oncology. However, resistance often comes at a fitness cost for the resistant cells [1,2]. Pest managers have long ago understood that pesticide resistance also comes at a fitness cost [3]. Pests that are sensitive to pesticide can outcompete pests that are resistant in the absence of or at low doses of pesticide. Pest managers have learned to use the minimum effective dose (MED) of pesticides rather than the MTD [3]. Current cancer treatments are aimed at killing as many tumor cells as possible in the shortest time, which is accomplished by utilizing drugs at the maximum tolerated dose (MTD). This approach is highly toxic and often results in a selective advantage for therapy-resistant cells, which eventually kills the patient (Figure 1A) [4,5]. Prior to treatment, a tumor is assumed to be mainly composed of drug-sensitive cells, with only a few pre-existing resistant cells. Treatment rapidly kills most sensitive cells, resulting in a competitive release of the resistant cells. In fact, evolutionary theory predicts that the fastest way to select for therapeutic resistance is to use the MTD [6]. Following this principle, we expect that in the absence of therapy, sensitive cancer cells will probably proliferate at the expense of the less-fit resistant cells. Thus, according to integrated pest management (IPM), a therapeutic strategy that is explicitly designed to maintain therapeutically sensitive organisms could increase a patient’s survival by using sensitive cells to suppress the growth of resistant cells. This is the inspiration for adaptive therapy in cancer treatment [4,5].

Adaptive therapy exploits intratumor cell competition to prolong patient survival by allowing sensitive cells to compete with resistant cells during periods of drug holiday or low drug dose [2,8]. Initial tests of adaptive therapy have been conducted with single drugs [2,4,8]. Two protocols have been tested in preclinical models: dose modulation (also known as AT1) [2,4,8] and dose skipping (also known as AT2) [2,4,8]. The dose modulation protocol compares the current size of the tumor to the previous measurement and then raises the drug dose if the tumor has grown (>20%), lowers the dose if the tumor has shrunk (>20%), and keeps the dose the same if the tumor burden has stayed within 80–120% (Figure 1B). The dose skipping protocol uses the MTD dose but skips a dose if the tumor has shrunk or remained stable since the last measurement. Results from preclinical models have shown that adaptive therapy with dose modulation was able to achieve indefinite control in mouse xenografts of metastatic triple-negative (MDA-MD-231) and estrogen-receptor-positive (ER+) (MCF7) breast cancer [2,4,8] as well as xenografts of ovarian cancer (OVCAR3) [4]. In contrast, the dose skipping (drug holiday) protocol was unable to prevent tumors from growing compared to standard of care [2,4,8]. On/off regimens are generally ineffective for fast-growing tumors due to a loss of control during the off cycle, but researchers have found application of this method in the context of slow-growing tumors, leading to their implementation in a clinical trial for prostate cancer [2,9,10].

The first clinical trial of single-drug adaptive therapy used a different protocol on metastatic, castration-resistant prostate cancer. This intermittent adaptive therapy protocol (Figure 1C) used MTD of single-agent abiraterone until the tumor burden (measured by prostate-specific antigen (PSA) in the blood) fell below 50% of its initial level. At that point, treatment was stopped and only resumed if the PSA level returned to its initial level. The administration of abiraterone was determined using mathematical models informed by evolutionary principles in order to prolong the emergence of resistance. This resulted in a doubling of time to progression using less than half of the cumulative drug dose compared to standard of care [8,9,10].

Our goal was to test if adaptive therapy can improve clinical outcomes, such as time to progression, overall survival, and reduction in toxicity, in breast cancer and, more specifically, to test multidrug adaptive therapies as well as single-drug therapy on endocrine-resistant breast cancer. Although a variety of adaptive therapy protocols have been tested in mice [2,4,8,11] and computational simulations [12,13,14], it remains an open question as to which version of adaptive therapy is best and under which conditions one protocol might be better than another [15]. Here, we tested different adaptive therapy protocols and compared them to the time to progression (TTP) of MTD in a preclinical model of breast cancer. We focused on endocrine-resistant ER+ breast cancer, which is an ongoing major clinical challenge.

We conducted a preclinical experiment to study the effect of gemcitabine and capecitabine on a xenograft model of endocrine-resistant ER+ cells in NOD SCID gamma (NSG) mice. As potential long-term treatments, the ideal drugs for adaptive therapy should have little cumulative toxicity and primarily affect proliferating cells. Many anticancer drugs that are classified as cell cycle specific (e.g., antitumor antibiotics) actually affect both proliferating and quiescent cells but only kill cells once they come out of quiescence. To reduce the effect on normal cells, we focused on antimetabolites that only affect proliferating cells and are used in breast cancer treatment, namely, gemcitabine [16] and capecitabine [17], both of which interfere with the synthesis of DNA. Our primary goal was to compare gemcitabine versus capecitabine for single-drug adaptive therapy. We compared gemcitabine to capecitabine to help inform the choice of drug for a future clinical trial. Furthermore, because we were interested in how multiple drugs might be combined in adaptive therapy, based on previous simulation results [15], we tested four different multidrug adaptive therapy protocols. Although the combination of gemcitabine and capecitabine is not theoretically justified as they have the similar mechanisms of action, there are some studies that suggest little cross-resistance between gemcitabine and capecitabine [18,19] (Appendix A). We capitalized on the opportunity to test multidrug adaptive therapies for the first time while also addressing the primary goal with single-drug adaptive therapy. These studies provide a framework for the rational design of adaptive combination chemotherapy protocols to control therapeutic resistance in breast cancer.

## 2. Materials and Methods

### 2.1. Study Design

The aim of this study was to test an evolution-based strategy to treat ER+ endocrine-resistant breast cancer. We first evolved the estrogen-positive (ER+) breast cancer cell line MCF7 to be resistant to fulvestrant (a selective estrogen receptor modulator (SERM), Selleck Chemicals LLC, Houston, TX, USA) and palbociclib (a CDK4/6 inhibitor, Selleck Chemicals LLC, Houston, TX, USA), both of which are commonly used clinically for ER+ breast cancer. We then tested a variety of adaptive therapy protocols and compared them to constant MTD therapy as well as vehicle control (no therapy), measuring time until death as the primary outcome. In all cohorts, the mice were randomly assigned to the treatment and control groups presented in Table 1. The primary endpoint for the mouse experiments was time to death (see below). We monitored tumor burden by the Xenogen IVIS Spectrum (PerkinElmer, Waltham, MA, USA) in vivo imaging system.

### 2.2. In Vitro Experiments

#### Cell Culture

##### Preparing Endocrine-Resistant Cell Lines

Human bioluminescent human ER+ breast cancer cell line able to express firefly luciferase (MCF7/luc) was cultured in RPMI 1640 (Gibco, ThermoFisher, Grand Island, NY, USA) and supplemented with 10% fetal bovine serum (FBS, (Gibco, ThermoFisher)) and 1% penicillin–streptomycin, an antibiotic (Gibco, ThermoFisher). MCF7 human breast cancer cell lines were provided by ATCC, then tagged with luciferase by the Gatenby lab at the Moffitt Cancer Center, which then shipped them to us. In order to select for resistance to fulvestrant and palbociclib, 1 × 10^7^ MCF7/luc cells from passage 3 were inoculated into a hyperflask and grown for 1 week with RPMI 1640 medium (500 mL) + 10% FBS until they reached approximately 1 × 10^9^ cells. We then treated them for one month with 70% inhibitory concentration (IC70, 14.8 μM) of the drugs for the first two weeks, IC80 for the third week (16 μM), and IC90 for the last week (18 μM). We started with a high dose to model the clinic and moved to even higher doses in order to select for high levels of resistance. We changed the medium of the hyperflask once per week when we added the drugs to the medium. When changing the medium, it was poured into an empty bottle by holding the hyperflask at an angle followed by addition of 100 mL of PBS to wash the flask. Finally, the fresh medium with drugs was poured into the flask.

##### Drug Dose–Response Curve Analysis of Resistant Cell Lines

We analyzed the Drug dose–response curve (DDR) on MCF7/luc cells that evolved to be resistant to fulvestrant and palbociclib versus the parental cell line (sensitive MCF7/luc cells) to compare their sensitivity to the drugs. We cultured the cells in Petri dishes with complete medium (RPMI 1640 medium +10% FBS), then harvested the cell using Accutase (VWR, Radnor, PA, USA) to detach the cells, followed by incubation in the incubator until the cells detached (about 5–10 min). Once the cells detached, equal amounts of complete medium were added, washed, and pipetted to bring all the cells into suspension. The cells were transferred into a tube and then counted, followed by centrifugation for 5 min at 1100 rpm. The medium was then removed, and the pellet was resuspended in a suitable volume for plating cells in order to obtain 10^3^ cells per well.

We used a 96-well plate with 2 controls (medium without drug and medium plus the solvent of each drug, in which we used the same percentage of solvent that we used for the highest concentration of each drug) and 10 treatments (drugs with different concentrations: 30, 40, 50, 60, 70, 80, 90, and 100 μM), with 8 replicates per treatment. Equal amounts of cells were plated on all wells on Day 1. After 24 h, the cells were treated with the drugs at the specified concentrations. The cells were incubated under the drugs for 72 h to cover at least one doubling time of the cell lines (MCF7S cells had a doubling time of 41 h, and MCF7R cells had a doubling time of 48 h). At the end of the experiment, an equal amount of 100 uL of CellTitre-Glo (Promega, Madison, WI, USA) was added to each well, followed by incubation for five minutes. CellTitre-Glo releases ATP from live cells [20], so the quantified luminescence data using a plate reader (SpectraMax M5 with the SoftMax Pro 6.2.2 software) correspond to live cells under each condition. These values were then normalized to construct a drug dose–response curve (DDR). Statistical analysis was performed on DDR results using GraphPad Prism. First, we log-transformed the concentration and then used a nonlinear regression with the following command: “(log(inhibitor) vs. response-variable slope (four parameters))”, which gave us a corrected DDR curve and an IC50 value.

### 2.3. In Vivo Experiment 

#### 2.3.1. Xenograft Model of Human Breast Cancer

We orthotopically implanted endocrine-resistant MCF7 cell lines tagged with firefly luciferase into the mammary fat pads of 8-week-old NOD/SCID gamma (NSG) mice. We used 3 × 10^6^ cells/100 uL for each mouse (suspended in DPBS/Matrigel 1:1). We used 6 mice per treatment group and 4 mice for the control group based on a power analysis: balanced one-way analysis of variance power calculation for 11 experimental arms and controls (groups = 11, n = 6, f = 0.53, power = 0.8, sig. level = 0.05). A total of 70 mice were used for this study. Orthotopic mouse xenograft experimental protocols were approved by the Institutional Animal Care and Use Committee (IACUC) at Arizona State University. NSG mice are immunodeficient, so they were maintained and evaluated under pathogen-free conditions in accordance with IACUC standards of care at the ASU vivarium.

For harvesting 3 × 10^6^ cells/mouse, the cells were first washed with Dulbecco’s phosphate-buffered saline (DPBS; Corning^TM^, Somerville, MA, USA) for 15 min to remove any trace of FBS from the cells. Then, the cells were detached using Accutase incubated at 37 °C, 5% CO_2_, for 5–10 min. Then, the cells were suspended in RPMI 1640 growth medium +10% FBS to neutralize the Accutase followed by centrifugation for 5 min at 1100 RPM. The pellet of cells was resuspended and mixed with 1:1 matrigel/PBS (3 × 10^6^ cells/100 μL). Matrigel was used to augment the tumor growth for our cancer cell lines [21].

For the MCF7 xenograft model, 17b-estradiol 90-day release pellets, 0.36 mg per pellet (Innovative Research of America), were implanted on the dorsal region of the mice on the day of cell injections in order to grow human breast cancer cells that are estrogen receptor positive [22,23]. The incision was closed with surgical glue, and the mice were monitored and kept warm while recovering from the anesthesia. They were returned to their cage once they recovered. This was repeated every 90 days while the mice were under observation.

#### 2.3.2. Tumor Burden Measurement

Tumor burden was measured by caliper and bioluminescence imaging under isoflurane anesthesia twice a week. The tumor volume measured by the caliper was calculated according to the following formula: volume = π (short diameter^2^) × (long diameter)/6. Imaging of live mice using the IVIS^®^ Spectrum (PerkinElmer, Waltham, MA, USA) revealed tumor size and density. For live imaging, we injected 100 μL of diluted XenoLight D-luciferin-K + salt, a bioluminescence substrate (PerkinElmer, Inc., Shelton, CT, USA), intraperitoneally at a concentration of 150 mg/kg body weight. Then, mice were anesthetized with isoflurane (3% induction dose and 1.5–2% maintenance dose via a precision vaporizer), followed by being placed in the chamber for fluorescent imaging using IVIS Spectrum (PerkinElmer, Inc.) and imaged ventrally. We found that there were a number of times when the caliper measures diverged from the bioluminescence measures of tumor burden (*r* = −0.019, *p* = 0.4). In some cases, the caliper measures indicated a large tumor but bioluminescence measured a low number of cancer cells. Upon histological examination, these tumors were often filled with fat. In other cases, there was no measurable tumor by calipers, but bioluminescence indicated significant cancer burden, either because the cancer cells were diffuse, invading the body cavity, or metastatic. Due to these issues, we chose to manage therapy based on bioluminescence measures as the most accurate indication of the cancer burden. However, because caliper measures were part of the approved IACUC protocol, mice were sacrificed if their tumor measured at least 2000 mm^3^ by calipers.

#### 2.3.3. Starting Therapy and Drug Dosing 

When the measured tumor volume exceeded 200 mm^3^ according to calipers or 3 × 10^8^ (photons/s/cm^2^/sr) as measured by bioluminescence imaging, mice were randomly divided into the control group and treatment groups. In treatment groups, we used gemcitabine (MedChemExpress, NJ, USA) at 50 mg/kg as the maximum tolerated dose (MTD), which we injected intraperitoneally twice weekly, and capecitabine given at 40 mg/kg/day as MTD, which we administered orally with a 20 uL pipette tip five days (Monday–Friday) per week. For the control group (no treatment), we injected saline (the solvent for both gemcitabine and capecitabine (Sigma-Aldrich, St. Louis, MO, USA)) twice a week (as a control for the twice-weekly injection of gemcitabine) and applied saline orally twice a week (as a control for the thrice-weekly oral application of capecitabine).

##### Standard Therapy Protocol

To model standard therapy, we applied the MTD of a drug continuously: 40 mg/kg/day of capecitabine five days a week or 50 mg/kg of gemcitabine twice a week. In the combined standard therapy, mice were given both drugs at MTD on those same schedules (gemcitabine twice a week and capecitabine five days a week).

##### Dose Modulation Adaptive Therapy Protocol

We adjusted drug dosage based on the tumor burden twice per week and optimized our experimental protocol according to the dose modulation strategy found in our agent-based modeling work [15]. Specifically, if tumor burden decreased by >10% since the last measurement based on bioluminescence, we decreased the dose by 50%. If the tumor burden increased by >10%, we increased the dose by 50% but not exceeding the MTD (40 mg/kg/day for capecitabine and 50 mg/kg for gemcitabine). When the tumor shrank below 2.25 × 10^8^ (photons/s/cm^2^/sr) or a minimum volume cutoff of 150 mm^3^ based on caliper measurements, we skipped the treatment until the tumor grew back to 3 × 10^8^ (photons/sec/cm^2^/sr).

##### Intermittent Adaptive Therapy Protocol

In this protocol, we started with the maximum tolerated dose of the drug: 40 (mg/kg/day) of capecitabine or 50 (mg/kg/2 days in a week) of gemcitabine. If the tumor burden fell below 50% of its value at the start of treatment, we withdrew the treatment. If it rose above the initial tumor burden at the beginning of treatment, we started the treatment again using the MTD of each drug. 

##### Ping-Pong Dose Modulation Adaptive Therapy Protocol

We used the dose modulation protocol as described above, except that we alternated between applying gemcitabine and capecitabine as follows:

Monday: Dose with gemcitabine.

Tuesday: Measure tumor burden and set the dose for the next application of gemcitabine (the following Friday) based on the tumor burden difference since last Friday. Start dosing with capecitabine.

Wednesday: Dose with capecitabine.

Thursday: Dose with capecitabine.

Friday: Measure tumor burden and set the dose for the next application of capecitabine on the following Tuesday based on the change in tumor burden since Tuesday.

Dose with gemcitabine.

Weekend: No treatment.

The dose was adjusted as described in the dose modulation protocol, adjusting the dose of a drug depending on whether the tumor grew or shrank the last time that drug was applied. Note that gemcitabine has a half-life of less than 1 h in mice, and capecitabine has a half-life of 1–4 h in mice [24].

##### Ping-Pong Intermittent Adaptive Therapy Protocol

We used one drug at a time, and like the single-drug intermittent adaptive therapy protocol above, we stopped treatment when the tumor burden fell below 50%. The only difference was that when the tumor burden returned to its initial value prior to treatment, we switched drugs rather than continuing with the same drug.

##### Tandem Dose Modulation Adaptive Therapy Protocol

This protocol was the same as the dose modulation adaptive therapy protocol above, except that both drugs were given and modulated in tandem (gemcitabine twice a week and capecitabine five days a week on weekdays). We measured tumor burden twice a week, and when the tumor grew more than 10% since the last measurement, we increased the dose of both drugs by 50% (up to but never going above their MTD). Similarly, if the tumor burden shrank by more than 10%, we decreased the dose of both drugs by 50%. 

##### Tandem Intermittent Adaptive Therapy Protocol

This was the same as the intermittent adaptive therapy protocol, except that both drugs were given at the same time. Both drugs were also withdrawn if the tumor burden fell below 50% of its initial burden and restarted if the tumor burden ever grew above 100% of its initial burden.

##### Endpoints

The primary endpoint for the mouse experiments was time to death. Mice were monitored daily and euthanized if they developed excessive lethargy; inappetence; excessive tumor burden; or other serious clinical conditions as determined by the veterinary staff, such as difficult or labored breathing, tumor interfering with locomotor activity, or reduced ability to obtain food or water. Excessive tumor burden was defined as 3 × 10^9^ (photons/s/cm^2^/sr) according to the luminescence data from IVIS or 2000 mm^3^ based on the calipers. Mice that had to be sacrificed due to the tumor interfering with locomotion or that died during the surgery to implant estrogen pellets were coded as censored for survival as long as the caliper measurements of tumor burden were below the threshold for sacrifice. All other reasons for sacrifice were coded as a mortality event, including sacrifice for excessive lethargy; inappetence; excessive tumor burden; or other serious clinical conditions as determined by the veterinary staff such as difficult or labored breathing, hunching, or reduced ability to obtain food or water (Appendix A). 

### 2.4. Ex Vivo Experiments 

#### 2.4.1. Derivation of Cancer Cell Lines from Mice 

When mice were euthanized, we harvested the tumors. Half of the tumor was used for deriving a cell line, and the other half was saved for tissue-based assays. To derive a cell line, the tumor was sliced and diced into small pieces. Then, 7 mL of the digest medium (6 mL collagenase (Burlington, Burlington, NJ, USA, 3 mg/mL PBS) + 10 mL TrypLE Express (Gibco, ThermoFisher, Grand Island, NY, USA) + 34 mL 1X sterile PBS) was added to the plate and incubated for 30 min at 37 °C. After incubation, the digest medium and cells were separated from any remaining tissue and transferred into a conical tube. Then, another 3 mL of digest medium was added to the remaining tissue and sliced again, followed by addition of 7 mL of digest medium and incubation of the plate in a tissue culture incubator for another 30 min. Next, we collected the medium and cells from the dish and added them to the same conical tube we had. Finally, we plated the cells in a Petri dish with a complete medium that contained Normocin (InvivoGen, San Diego, CA, USA) at a 1:500 ratio and put them in the incubator to grow. Then, the cell lines were frozen and kept at −80 °C.

#### 2.4.2. Histological Analysis and Immunohistochemistry

The samples were embedded in a paraffin block for hematoxylin and eosin (H&E) staining and immunohistochemistry (IHC). Tumor tissue was sectioned in 6 μm slices using Leica Ultracut-R Microtome. We performed IHC using the following antibodies: Ki67, which was used as a marker of proliferation to assess the proliferative activity of the tumor and to determine tumor response to our treatment protocols (Ki-67 recombinant rabbit monoclonal antibody (SP6), Invitrogen #MA5-14520) [25], and caspase 3 (Cleaved Caspase-3 (Asp175) Antibody #9661, Cell Signaling) [26,27], which is an important component of apoptosis (Appendix A). 

### 2.5. Statistical Methods

We analyzed the survival results using Kaplan–Meier analyses and Cox proportional hazard models. We analyzed our histology and IHC comparing MTD and adaptive therapy groups with the Mann–Whitney test.

### 2.6. Computational Modeling

We modified our previously published hybrid-agent-based model [15], which is an extension to the Hybrid Automata Library (HAL) agent-based modeling framework [28], to simulate adaptive therapy using a single cytotoxic or two cytotoxic drugs. The description of the model can be found in [15], with the following changes.

As noted in Section 2.2.2 (Entities, State Variables, and Scales), for treatment with a single cytotoxic drug, we considered two different cell types: sensitive and resistant. As noted in Section 2.4.11 (Observation), we modified our criterion for progression. At any point after the initiation of therapy, if the tumor burden equaled or exceeded 99% of the carrying capacity or if the rolling average of the resistant cells (doubly resistant for treatment with two drugs or singly resistant for treatment with one drug) over 500 time-steps equaled or exceeded 50% of the carrying capacity, the particular run was scored as “progressed”. As noted in Section 2.5 (Initialization), we considered two different cell types: sensitive and resistant. As noted in Section 2.7.1 (Cell Death), for treatment with a single cytotoxic drug, the equation for the probability of cell death was as follows: Probability of cell death per hour = background death probability per hour + S1 × [Drug1] × Ψ1, where S1 is the binary indicator variable for the cell’s sensitivity to drug 1, [Drug1] is the concentration of drug 1 (non-negative real values), and Ψ1 is the drug potency (non-negative real values) quantified as the probability of cell death per unit drug concentration per hour. As noted in Section 2.7.2 (Cell Division), the cell division rate for the sensitive cells was 0.06 per hour, and the cell division rate for the resistant cells was 0.02 per hour. As noted in Section 2.7.4 (Mutation), for treatment with a single cytotoxic drug, the default value for the mutation rate parameter was 1 × 10^−3^ per cell division, accounting for the transition from sensitive to resistant cell types and vice versa. Protocols for treatment with a single cytotoxic drug and two cytotoxic drugs are described in Appendix A. The parameter values used to run the model for treatment using a single cytotoxic drug are shown in Table 2A, and those used to run the model for treatment using two cytotoxic drugs are indicated in Table 2B. MTD dosage value for treatment using a single drug was set to 2.5 units, and MTD dosage value for treatment using two cytotoxic drugs was set to 2.5 units for each of the two drugs; thus, the total dosage equaled 5 units for a cocktail application and 2.5 units of the specific drug for the ping-pong protocols. This matched the mouse experiments, in which the dosage of gemcitabine and capecitabine were not lowered when they were used in combination.

## 3. Results

We tested whether adaptive therapy works in preclinical mouse models with endocrine-resistant breast cancer and compared adaptive therapy with standard therapy. We tested the two leading protocols for adaptive therapy (dose modulation vs. intermittent) in mice using gemcitabine and/or capecitabine on the human breast cancer cell line MCF7 evolved to be resistant to fulvestrant and palbociclib. 

### 3.1. Endocrine-Resistant MCF7 Cell Line

Typically, it takes 6–18 months to select for therapeutically resistant cells in a cell line [29]. However, resistance often emerges in the clinic in a matter of a few months [30,31,32]. We hypothesized that this discrepancy may be due to the use of unrealistically small cell population sizes in vitro. Population size is a key parameter of the rate of evolution [33]. Using a hyperflask to culture a population of ~10^8^ MCF7 cells, we were able to generate cells that were resistant to five-fold higher doses of fulvestrant and palbociclib in one month (Figure 2A). After a month of exposure to both fulvestrant and palbociclib, MCF7 resistant (MCF7 R) cells had evolved a half-maximal concentration inhibitory concentration (IC50) value of 62.65 μM compared to 13.13 μM for the sensitive parental cells (MCF7 S). Moreover, we tested gemcitabine and capecitabine on both of the sensitive and resistant cells (Figure 2B). We found MCF7 sensitive cell lines showed IC50 values of 153 and 235 μM to gemcitabine and capecitabine, respectively. Gemcitabine and capecitabine showed IC50 values of 134 and 128 μM on MCF7 resistant cell lines (resistant to fulvestrant and palbociclib), respectively.

### 3.2. Tumor Growth Control after Therapy Cessation

In 20 of the 48 mice on adaptive therapy protocols, the tumor burden shrank below our minimum threshold at which we stopped therapy and then remained at low tumor burden for many weeks until the mice died or had to be sacrificed for other reasons (noted in Appendix A as “tumor remained stable after dosing stopped”). It is not clear why tumor cells still detectable by bioluminescence did not grow in the absence of drugs or a functional immune system, though in half of the cases, the tumor size did grow as measured by calipers (Appendix A). In an additional seven mice, the tumor stayed stable after the cessation of therapy but eventually did regrow.

### 3.3. Prolonged Survival Benefit of Adaptive Therapy

#### 3.3.1. Single-Drug Therapy

In capecitabine, single-drug adaptive therapy (Figure 3) survival was better, though not statistically significant, for dose modulation compared to no treatment (Cox proportional hazards HR = 0.24, 95% CI = 0.042–1.4, *p* = 0.1) and compared to MTD (HR = 0.22, 95% CI = 0.043–1.1, *p* = 0.065). There was some evidence that dose modulation prolonged survival compared to intermittent therapy (Figure 3, HR = 0.26, 95% CI = 0.049–1.4, *p* = 0.09). There were no differences between intermittent therapy relative to no treatment (HR = 0.96, 95% CI = 0.247–3.7, *p* = 0.95) and MTD (HR = 0.78, 95% CI = 0.236–2.6, *p* = 0.68). There was no significant benefit in prolonging survival between MTD therapy relative to no treatment (HR = 1.2, 95% CI = 0.323–4.5, *p* = 0.78) (Figure 3A). In addition, a lower average total drug dose was used in dose modulation (533.8 mg/kg) compared to intermittent (1226.6 mg/kg) and MTD (1373.3 mg/kg) therapies (Appendix A). In gemcitabine single-drug adaptive therapy, there were no statistically significant differences in survival time for any of the protocols (Figure 3B). However, we found that a lower average total drug dose was used in dose modulation (256.9 mg/kg) compared to intermittent (333.3 mg/kg) and MTD (783.3 mg/kg) therapies (Appendix A).

#### 3.3.2. Multidrug Therapy

When using both gemcitabine and capecitabine, survival was better for ping-pong intermittent treatment relative to no treatment (HR = 0.13, 0.13, 95% CI = 0.02–0.84, *p* = 0.032). There was evidence that survival was also better for ping-pong dose modulation compared to no treatment, but it was not quite statistically significant (Cox proportional HR = 0.2, 95% CI = 0.038–1.09, *p* = 0.06). In comparison to MTD, both ping-pong dose modulation and ping-pong intermittent protocols prolonged survival (dose modulation: HR = 0.11, 95% CI = 0.024–0.55, *p* = 0.007; intermittent therapy: HR = 0.07, 95% CI = 0.013–0.42, *p* = 0.003). Survival under MTD with both drugs was worse than no treatment but not significantly (HR = 1.9, 95% CI = 0.48–7.4, *p* = 0.36) (Figure 4A). Moreover, the average amount of drugs used per day was lower for both AT therapies (GEM: 4.9 mg/kg, CAP: 6.1 mg/kg per day for ping-pong dose modulation and GEM: 6.4 mg/kg, CAP: 8 mg/kg per day for ping-pong intermittent) compared to MTD (GEM:20.6 mg/kg, CAP: 30.6 mg/kg per day) (Appendix A).

With tandem combination adaptive therapy, where both drugs were used at the same time, dose modulation and intermittent protocols were better than combined MTD therapy, but the improvement was only statistically significant for dose modulation (Cox proportional hazards for tandem dose modulation vs. MTD: HR = 0.22, 95% CI = 0.05–0.94, *p* = 0.04; for intermittent vs. MTD: HR = 0.34, 95% CI = 0.07–1.6, *p* = 0.17). Neither form of tandem adaptive therapy was significantly better than no treatment (tandem dose modulation relative to no treatment: HR = 0.34, 95% CI = 0.07–1.6, *p* = 0.17; tandem intermittent relative to no treatment: HR = 0.73, 95% CI = 0.17–3, *p* = 0.66). MTD was worse than no treatment in survival, but the difference was not statistically significant (HR = 1.9, 95% CI = 0.48–7.4, *p* = 0.36) (Figure 4B).

### 3.4. A Strong Correlation between the Percentage of Maximum Tolerated Drug Dose Used with Survival Time

We calculated the cumulative total drug dose administered and divided it by the number of days between the start of therapy and death to get the average drug dose per day, normalized by the MTD. We found a strong negative correlation between the %MTD drug dose per day and the survival time of mice in combination therapy (*p* < 0.0001, Figure 5A). We also observed a significant negative correlation between the %MTD drug dose per day and the survival time of mice in single-drug therapies (capecitabine or gemcitabine) (*p* = 0.0074, Figure 5B). There was no relationship between the final tumor burden size and the survival time (Appendix A), though there was a trend of higher concentrations of drug resulting in larger tumors at the end of the experiment (*p* = 0.076, Appendix A). 

### 3.5. Different Chemosensitivity in Cell Lines Retrieved from Different Treatment Groups

To determine the chemosensitivity of the retrieved cell lines from the tumors, we generated drug dose–response (DDR) curves for single-drug therapies and combination therapies (Appendix A). We generated DDR curves on parental pre-engrafted MCF7 breast cancer (ER+) cell lines with single and combination treatment of capecitabine and gemcitabine. Then, we compared the sensitivity of engrafted cell lines under different treatment strategies for each drug to pre-engrafted MCF7 cell lines treated with that drug. In single-drug therapy using gemcitabine and combination therapy, we found cell lines from MTD groups showed the highest IC50s values, with the apparent exception of intermittent therapy with capecitabine (Figure 6). However, the two cell lines from tumors that were treated with the capecitabine intermittent protocol both came from tumors that never fell below 50% of their initial tumor burden and so never had a drug holiday. Thus, they were treated the same as the MTD condition, which may explain why they resulted in relatively high IC50 values. 

In capecitabine single therapy, we were able to retrieve cell lines from three mice of the MTD group, two mice in the dose modulation group, two mice in the intermittent group, and three mice that were not treated. Figure 6A shows the capecitabine IC50 values for those cell lines (drug dose–response curves are shown in Appendix A). 

Gemcitabine IC50 values for mice treated with gemcitabine alone are shown in Figure 6B (drug dose–response curves are shown in Appendix A). For mice treated with the combination of both drugs, there were more dramatic differences in the IC50 values between conditions (Figure 6C). For the two cell lines from Mice 55 and 58 treated with the ping-pong dose modulation protocol and the one cell line from Mouse 50 treated with the intermittent tandem protocol, the cells were statistically significantly more sensitive to combination therapy compared to the cell line from Mouse 38 treated with MTD of both drugs (*t*-test *p* < 0.001 at 100, 150, and 250 μM comparing Mice 55 and 58 vs. 38; for Mouse 50 vs. 38, *p* = 0.0026 at 100 μM and *p* < 0.0001 at both 150 and 250 μM; Supplemental DDR curves in Appendix A). We were unable to derive a cell line from mice treated with the ping-pong intermittent protocol, so they are not included in Figure 6C.

### 3.6. Correlation between IC50 Values with Both Tumor Burden and Drug Dose

We hypothesized that a growing tumor at the time of death might indicate therapeutic resistance. We analyzed the relationship between the IC50 values of cell lines retrieved from the mice and the last measured tumor burden in those mice based on bioluminescence. We found a significant positive correlation (*p* = 0.0041, R^2^ = 0.43; Figure 7A). Moreover, there was a positive correlation between the average drug dose applied per day for each mouse and the resulting IC50 values of the cells derived from those tumors (*p* = 0.04, R^2^ = 0.25; Figure 7B), but this result is not statistically significant after correcting for multiple testing. 

### 3.7. Immunohistochemistry and Histological Analysis

We analyzed the percentage of apoptotic cells (caspase-3 positive) and percentage of proliferating tumor cells (Ki-67 positive) out of the total number of cells examined (a standard field of view was utilized to compare the proportion of positive cells) on our tumor sections for all the tumors (Appendix A). The maximum tolerated dose of both capecitabine and gemcitabine in single and combination strategies led to higher expression of Ki-67 than adaptive therapy protocols (*t*-test *p* = 0.0026; Figure 8A). In contrast, the maximum tolerated dose of both capecitabine and gemcitabine in single and combination strategies had lower caspase-3 index (lower apoptosis) compared to all the adaptive therapy groups (*t*-test *p* = 0.0457; Figure 8B), though this was not statistically significant after multiple testing corrections. There was no evidence of a correlation between proliferation and apoptotic indices in the tumors (Appendix A). We also assessed necrosis in tumors recovered from mice at the end of the experiment, scoring it on a scale from 0 to 4 (Appendix A).

### 3.8. Computational Simulations Match the Rank Orders for the Different Adaptive Therapy Protocols

We previously developed a hybrid-agent-based model to explore adaptive therapy protocols using one [34] or two [7] cytotoxic drugs. Although our model is simple and does not represent the details of breast cancer, the results of the simulations and mouse experiments using gemcitabine or capecitabine are consistent with respect to the rank order of their success. In both the modeling (Figure 9A) and experiments (Figure 9B) with capecitabine alone, the dose modulation protocol worked best, and the intermittent protocol was essentially equivalent to MTD. For the two-drug protocols, the dose modulation ping-pong protocol, the dose modulation tandem protocol, and the ping-pong intermittent protocol were all equivalent and better than the other protocols in both the simulation and experimental results (Figure 9). The standard MTD protocol and the tandem intermittent protocols are equivalent and worse than the other protocols in the mouse experiment, though in three of the six mice in the tandem intermittent group, the tumor never shrank below the threshold to stop dosing at MTD, so they actually received the same treatment as the MTD group (Appendix A). In the simulations, the tandem intermittent protocol did result in treatment holidays and did better than MTD but still not as well as the dose modulation and ping-pong protocols (Figure 9C). 

## 4. Discussion

Cancers are highly dynamic, and cancer cells evolve phenotypic strategies to deal with the challenges of therapy. This suggests that our treatments should also be dynamic and adjust to how the cancer is changing [2,4,8]. Standard therapy using MTD of cancer drugs imposes the strongest selective pressure to select for resistant phenotypes [35]. Typically, that eliminates sensitive populations, resulting in the competitive release of resistant cells and ultimately treatment failure [6]. Adaptive therapy was translated from integrated pest management as a strategy for preventing therapeutically resistant clones from growing out of control, thereby improving patient survival, with the added benefit of reducing toxicity. 

The innovation of this experiment was to test adaptive therapy with multiple drugs and testing the ping-pong and intermittent protocols for the first time in mice. It is also the first time that adaptive therapy has been tested on an ER+ but endocrine-resistant breast cancer cell line. One advantage of the ping-pong protocol is that only one drug is applied at a time, which should both reduce toxicity and limit selection for multidrug resistance. These strategies worked best in both our simulations [7] and experiments. In general, dose modulation adaptive therapies worked better than fixed (MTD) doses, probably because reducing the dose both reduced toxicity and selection for resistance (Figure 5 and Figure 7). Furthermore, in the intermittent protocol, we only stopped dosing if the tumor burden fell below 50% of its initial value. This means that if the tumor never fell below that threshold, we kept dosing at MTD. In those cases, there was no difference between the intermittent and MTD protocols. 

In many cases, adaptive therapy was able to maintain control over tumor growth with low doses. In 27 of the mice treated with adaptive therapy, the tumor burden fell below 2.25 × 10^8^ (photons/sec/cm^2^/sr) or 150 mm^3^, which triggered a cessation of treatment. In 20 of those 27, the tumor continued to shrink or stayed stable after therapy was stopped (Appendix A). This was a surprise. It may be due to some form of an Allee effect, in which small, fragmented cancer cell populations have difficulty regrowing [36,37,38]. Alternatively, it may be the result of neutrophils helping to control the tumors as NSG mice still have functional neutrophils [39]. But the mechanism of that effect has yet to be determined. We should also note that in three of the four control mice that were only treated with saline, the tumor stayed relatively stable over time (Appendix A).

By deriving cell lines from many of the tumors at the end of the experiment, we were able to test levels of therapeutic resistance that had evolved under the different protocols. We found that tumors exposed to long-term MTD tended to evolve more resistance than tumors treated with adaptive therapies (Figure 6) and grow large (Figure 7A). The resistance that evolved was correlated with the average dose applied per day (Figure 7B). It is unclear why, in a few cases, mice treated with a vehicle alone evolved tumors that were resistant to gemcitabine and/or capecitabine (Figure 6). The one mouse that evolved clear resistance to both drugs (Mouse 68) also evolved metastasis (though the cell line we tested for resistance came from the primary tumor). There may be a mechanistic association between the capacity to metastasize and resistance to antimetabolite drugs. Another disturbing possibility is that resistance to capecitabine and gemcitabine does not always come with a fitness cost. 

Evolutionary life history theory suggests that there is likely to be a tradeoff between maximizing survival versus maximizing proliferation in cancer cells [40]. Populations in stable environments tend to evolve slow life histories because they expand until resources are scarce and then have to compete for those limited resources [41]. We had hypothesized that adaptive therapy, by attempting to maintain a stable tumor size, might select for cancer cells with a slower life history strategy than MTD. There are two counterarguments to this hypothesis. The fact that we achieved stable population size through external mortality (cytotoxic drugs) rather than resource limitations might select for fast life history strategies. Furthermore, constant high dosing with cytotoxins might select for forms of resistance that invest heavily in survival over reproduction. In fact, adaptive therapy is based on the assumption that resistance has a fitness cost in the absence of drugs. However, when therapy is applied at high dose with recovery periods between doses, there may be selection for clones that can withstand therapy and then rapidly proliferate in the periods between therapy. Our results showed that tumors that evolved under constant high-dose treatment had high levels of proliferation and low levels of apoptosis despite being exposed to cytotoxic drugs. In contrast, tumors that evolved under adaptive therapies had low levels of proliferation and higher levels of apoptosis, which should lead to a slower net proliferation rate compared to MTD (Figure 8). Thus, it appears that adaptive therapy protocols selected for slower life history strategies.

We discovered a simple way to increase the speed with which we can evolve resistant cell lines in vitro based on evolutionary theory. Theory suggests that the rate of evolution is determined by five parameters: population size, generation time, mutation rate, selective coefficients, and the heritability of those selected phenotypes. We hypothesized that the reason acquired therapeutic resistance often evolves in the clinic in a matter of a few months while most in vitro protocols require 6–18 months, often with slowly escalating doses, is due to a difference in the population sizes. Clinical tumors are estimated to have 10^8^ cells per cm^3^ [42], whereas typical cell culture conditions often have only 10^5^ cells. We treated 10^8^ cells in a hyperflask at IC70 of both fulvestrant and palbociclib and were able to evolve a resistant variant of MCF7 cells in just one month.

### 4.1. Caveats

This study had some caveats that could be addressed in follow-up investigations. First, with only six mice per group, our statistical power was limited for detecting survival differences between the protocols. Second, we continued to dose at MTD in the control condition as a control for the possibility that adaptive therapy may succeed simply by continuing to dose. This matches clinical practice for capecitabine [43] but not gemcitabine, which is usually given for a limited period of time [44]. Gemcitabine given at MTD may have performed better if we had stopped dosing after a fixed number of applications. Furthermore, we only tested a single cell line. Adaptive therapy is likely to work differently on different tumors and with different drugs, particularly if there are differences in the cost of resistance. We used two drugs with similar, though not identical, mechanisms of action. Ideally, drugs that select for very different mechanisms of resistance should be used, which suggests we should choose drugs with very different mechanisms of action. It is reassuring that there may be little cross-resistance between gemcitabine and capecitabine [36,37,38]. Much of the advantage of adaptive therapies with multiple drugs that we observed was likely due to comparison with our highly toxic, constant “MTD” dose control condition. The fact that our multidrug “MTD” condition was worse than no treatment suggests that the doses we used for the combined gemcitabine and capecitabine caused a lot of toxicity and were actually above the maximum tolerable dose.

### 4.2. Future Challenges and Opportunities

The observation that dose modulation adaptive therapy with capecitabine appeared to extend time to progression and reduced toxicity compared to MTD may justify clinical trials in ER+ metastatic, endocrine-resistant but chemo-naive breast cancer. Furthermore, the success of ping-pong dose modulation adaptive therapy suggests that we should try interleaving different single-drug therapies to both reduce toxicity and to avoid selecting for multidrug resistance [15]. Future mouse experiments should test lower doses of capecitabine to reduce toxicity and also pair it with a drug with a very different mode of action. Furthermore, we should test ping-pong protocols alternating every cycle (used here) and ping-pong on progression [15], in which a single drug is used at a time and its dose reduced as long as the tumor is shrinking, with a switch to the other drug only conducted when the tumor starts to grow. Using dose modulation but waiting for progression before switching drugs may help clones shrink that are resistant to the drug not currently being used so that the tumor is more sensitive when we switch drugs. Our simulations suggest that this is a particularly effective two-drug adaptive therapy protocol [15].

In general, successful translation of adaptive therapy to the clinic will require drugs for which their dose can be easily modulated and have little to no cumulative toxicity. We will also need biomarkers that help distinguish between potentially curable cancers from those that already harbor resistant clones for which we should shift to the goal of control rather than cure. Even in cases where we do not know all the mechanisms of therapeutic resistance for a drug, measurements of intratumor heterogeneity may serve to identify cancers with high levels of clonal heterogeneity that likely harbor resistant clones.

For both adaptive therapy and other cancer therapy protocols, there is a critical clinical need for relatively inexpensive and noninvasive methods to measure tumor burden, which can be used frequently throughout treatment to monitor tumor response. It is difficult to manage and effectively respond to a dynamic, evolving disease if we only measure its response to therapy sometime after we have completed a fixed protocol. We predict that treatment protocols that are conditional on how the tumor is responding, such as adaptive therapy, will be more effective than fixed protocols, which are insensitive to how the tumor is responding. There is considerable variation in how different tumors respond to a given therapy, and protocols like adaptive therapy are able to individualize treatment based on those differences. Such individualized treatment will likely be a central component of future precision medicine.

## 5. Conclusions

Adaptive therapies show promise in this ER+ endocrine-resistant model of breast cancer. We found that mice under adaptive therapy treatments not only had better survival but their tumors also evolved a lower proliferation index and higher cell death rate compared to MTD protocols. In fact, across all the variations of protocols we tested, we found that the more drugs that were used, the shorter the survival time and the greater the resistance that evolved in the tumors. This implies that in the setting of late-stage cancers and second-line therapy or any other setting where cancers are not being treated with intent to cure, we may be able to extend time to progression by reducing the dose of anticancer drugs and perhaps even use evolutionary approaches to prevent the evolution of acquired therapeutic resistance. In the setting of endocrine-resistant breast cancer, our results also support clinicians’ preference of capecitabine over gemcitabine. The success of adaptive therapy here could provide justification for a future clinical trial using dose modulation adaptive therapy with capecitabine in ER+ metastatic, endocrine-resistant but chemo-naive breast cancer. If a second drug were added to capecitabine, we would recommend using them in a dose modulation, ping-pong on progression protocol in which the standard de-escalation of doses is used as long as the tumor is shrinking on a single drug but instead of raising the dose if the tumor grows, a switch is made to the other drug and de-escalation resumed until progression [15]. By changing the way we use current drugs so as to prevent the expansion of resistant clones, we have the opportunity to transform cancer from an acute, lethal disease into one that is chronic and manageable.

## Figures and Tables

**Figure 1 cancers-16-00257-f001:**
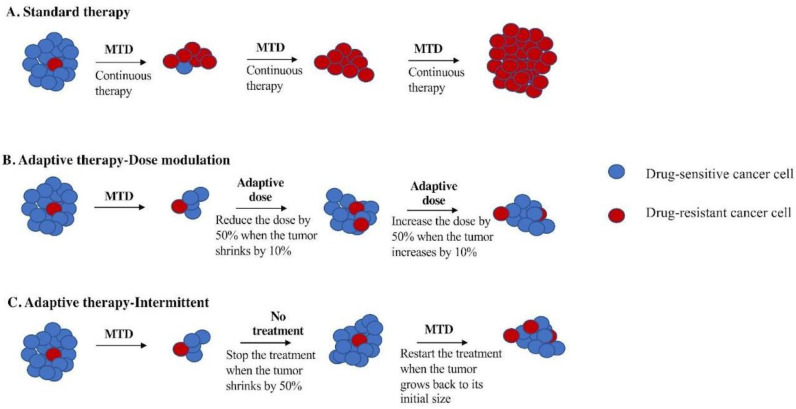
Schematic figure of comparing adaptive therapy protocols with standard therapy. (**A**) Standard therapy selects for cells (red) that are resistant to treatment and tumor relapse. Adaptive therapy maintains a stable tumor volume by preserving drug-sensitive cells (blue), suppressing the growth of less fit, resistant cells (red). (**B**) Dose modulation adaptive therapy raises the dose if the tumor grows and lowers the dose if the tumor shrinks. Previous mouse models have used a tumor burden change of 20% to trigger a change in dose [2]. Our simulation studies suggest a lower threshold is better [7], so we have used a 10% change in tumor burden to trigger a change in dose in this study. (**C**). Intermittent adaptive therapy stops dosing altogether if the tumor burden falls below a threshold (e.g., 50% of its initial value) and restarts treatment if the tumor recovers (e.g., to 100% of its initial value) [8,9].

**Figure 2 cancers-16-00257-f002:**
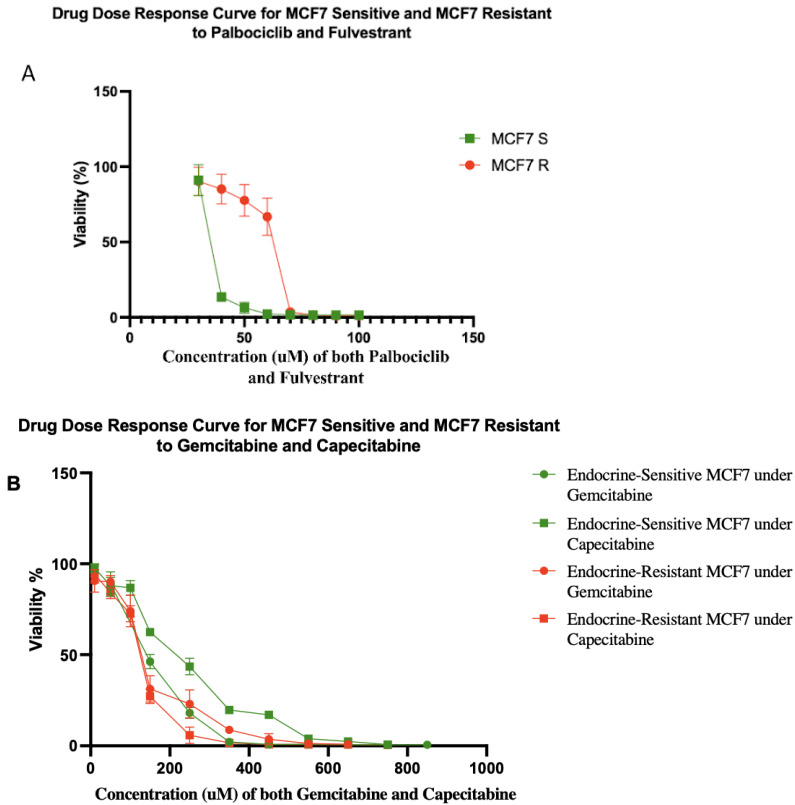
(**A**) Drug dose–response curves for sensitive and resistant MCF7/luc cell lines to combination of fulvestrant and palbociclib. MCF7 resistant (MCF7 R) and sensitive (MCF7 S) cell lines showed IC50 values of 62.65 and 13.13 μM, respectively. The percentages of viability at 40, 50, and 60 μM were significantly different between MCF7 R and MCF7 S cell lines. Each data point represents eight replicates. (**B**) Drug dose–response curves for sensitive and resistant MCF7/luc cell lines to gemcitabine and capecitabine. Green lines show drug dose–response of endocrine-sensitive MCF7 cell lines to gemcitabine with IC50 value of 153 μM and capecitabine with IC50 value of 235 μM. Red lines show drug dose–response of endocrine-resistant (resistant to fulvestrant and palbociclib) MCF7 cell lines to gemcitabine with IC50 value of 134 μM and capecitabine with IC50 value of 128 μM. Note that, if anything, resistance to endocrine therapy led to increased sensitivity to antimetabolite therapy in these cells.

**Figure 3 cancers-16-00257-f003:**
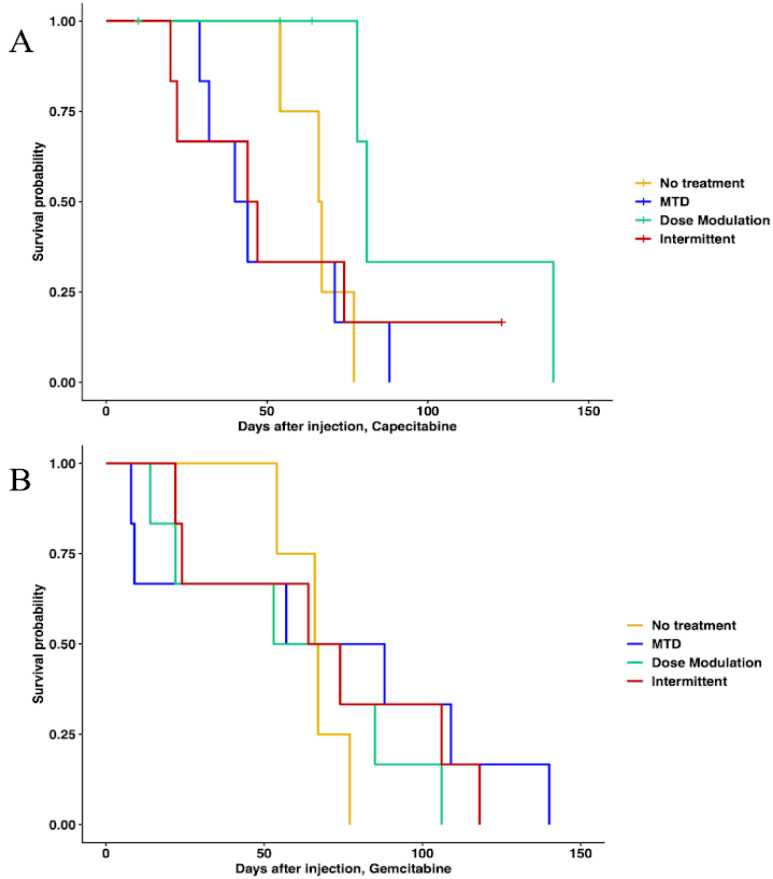
Survival analysis of single-drug therapies. (**A**) Survival analysis of capecitabine protocols using Cox regression (CAP dose modulation relative to no treatment: HR = 0.24, 95% CI = 0.042–1.4, *p* = 0.1; CAP intermittent relative to no treatment: HR = 0.96, 95% CI = 0.247–3.7, *p* = 0.95; CAP MTD relative to no treatment: HR = 1.2, 95% CI = 0.323–4.5, *p* = 0.78; CAP dose modulation relative to MTD: HR = 0.22, 95% CI = 0.043–1.1, *p* = 0.065; CAP intermittent relative to MTD: HR = 0.78, 95% CI = 0.236–2.6, *p* = 0.68). (**B**) Survival analysis of gemcitabine protocols. None of the protocols were statistically significantly different using Cox regressions.

**Figure 4 cancers-16-00257-f004:**
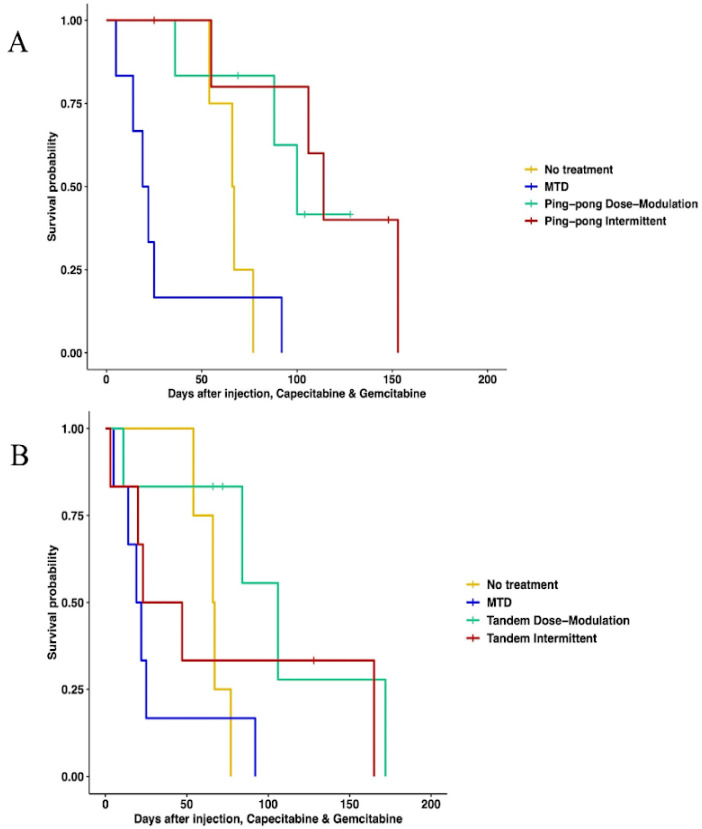
Survival analysis of multidrug therapies. (**A**) Multidrug ping-pong adaptive therapy protocols versus MTD or no treatment (vehicle control). In ping-pong protocols, only one drug is used at a time. Here, MTD is the application of MTD of both gemcitabine and capecitabine using the same dose scheduling as a single drug. (**B**) Multidrug tandem adaptive therapy protocols versus MTD or no treatment (vehicle control). In the tandem and MTD protocols, both gemcitabine and capecitabine were given (and modulated) at the same time.

**Figure 5 cancers-16-00257-f005:**
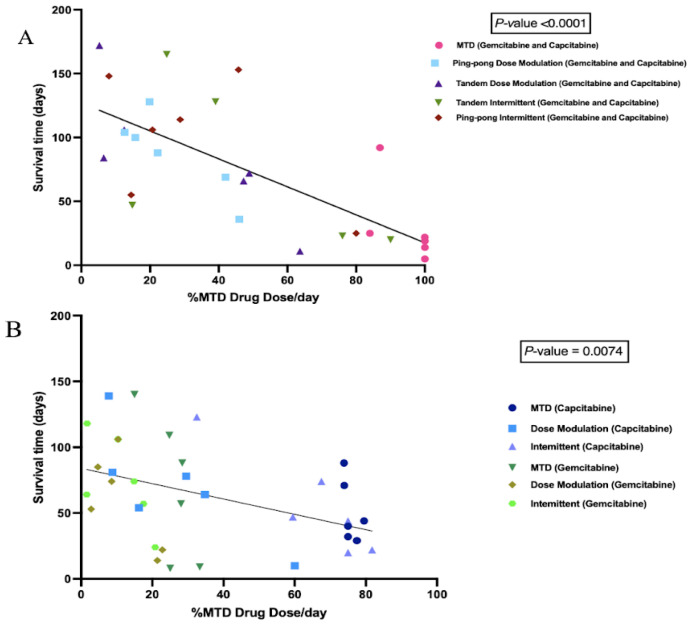
Correlation between the percentage of maximum tolerated drug dose used with survival time. (**A**) Survival time as a function of the amount of combined gemcitabine and capecitabine that was given per day. Mice that were given more chemotherapy tended to have a shorter survival time (*p* < 0.0001, R^2^ = 0.56). (**B**) Survival time as a function of the amount of single drug (either gemcitabine or capecitabine) that was given per day. Although there were a lot of variances associated with different protocols, there was still a strong trend that mice treated with more drugs had a shorter survival time (*p* = 0.0074, R^2^ = 0.19).

**Figure 6 cancers-16-00257-f006:**
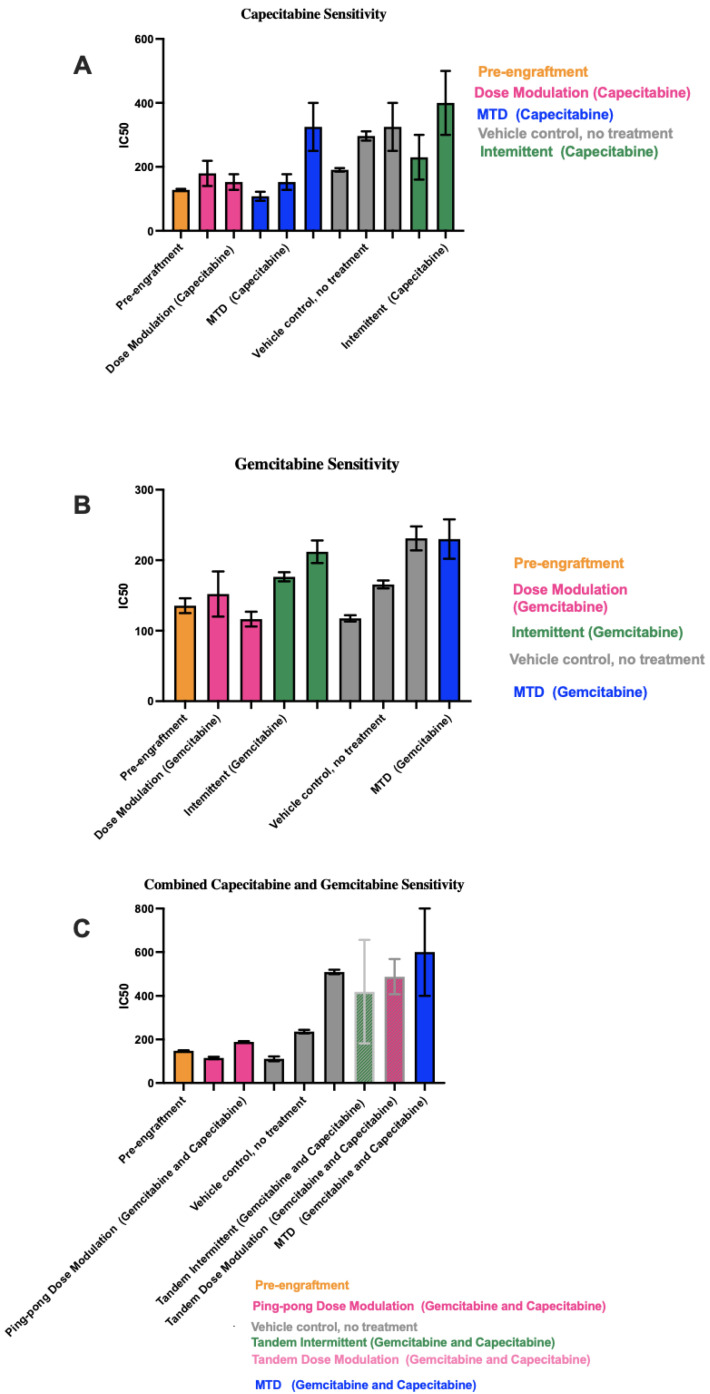
IC50 values for cell lines derived from tumors under different treatment conditions. In each panel, the protocols have been ordered from lowest to highest mean IC50 values across the 1–3 cell lines we were able to derive from the tumors for each condition. (**A**) Capecitabine IC50 values for mice treated with capecitabine alone. (**B**) Gemcitabine IC50 values for mice treated with gemcitabine alone. (**C**) Combined capecitabine and gemcitabine IC50 values for mice treated with both drugs together. Error bars show the 95% confidence intervals on the IC50 values based on eight replicates at each drug concentration in our drug dose–response experiments.

**Figure 7 cancers-16-00257-f007:**
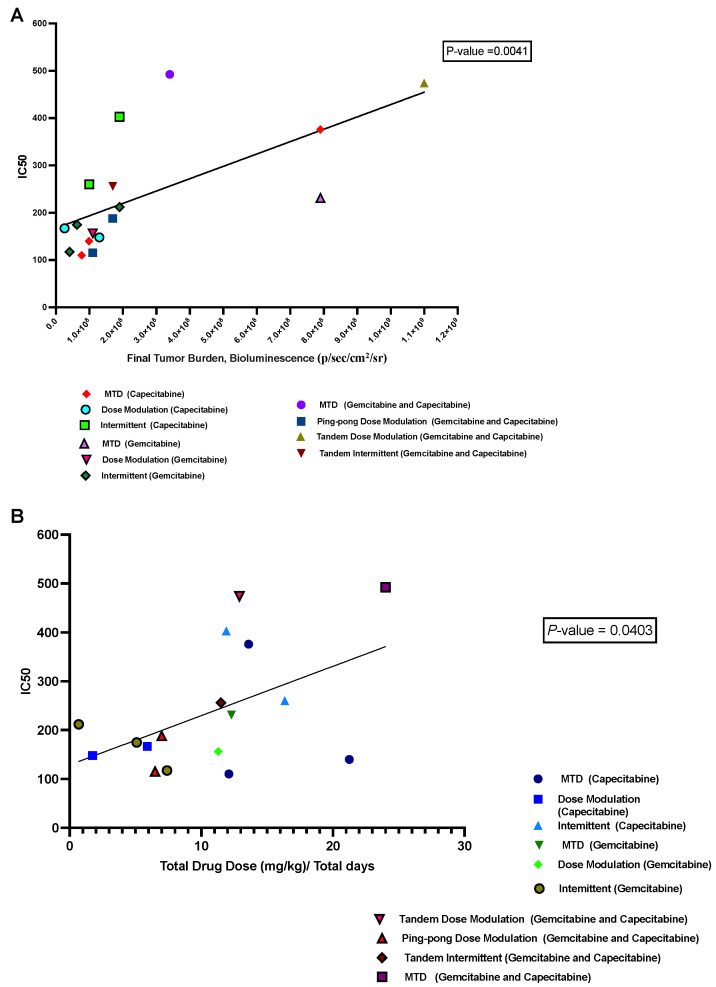
Correlation between IC50 values with both tumor burden and drug dose. (**A**) The relationship between the tumor burden at death and the IC50 values of the cell lines derived from those tumors. (**B**) The relationship between the average amount of drug used per day to treat a mouse and the resulting IC50 of the cells derived from that mouse’s tumor.

**Figure 8 cancers-16-00257-f008:**
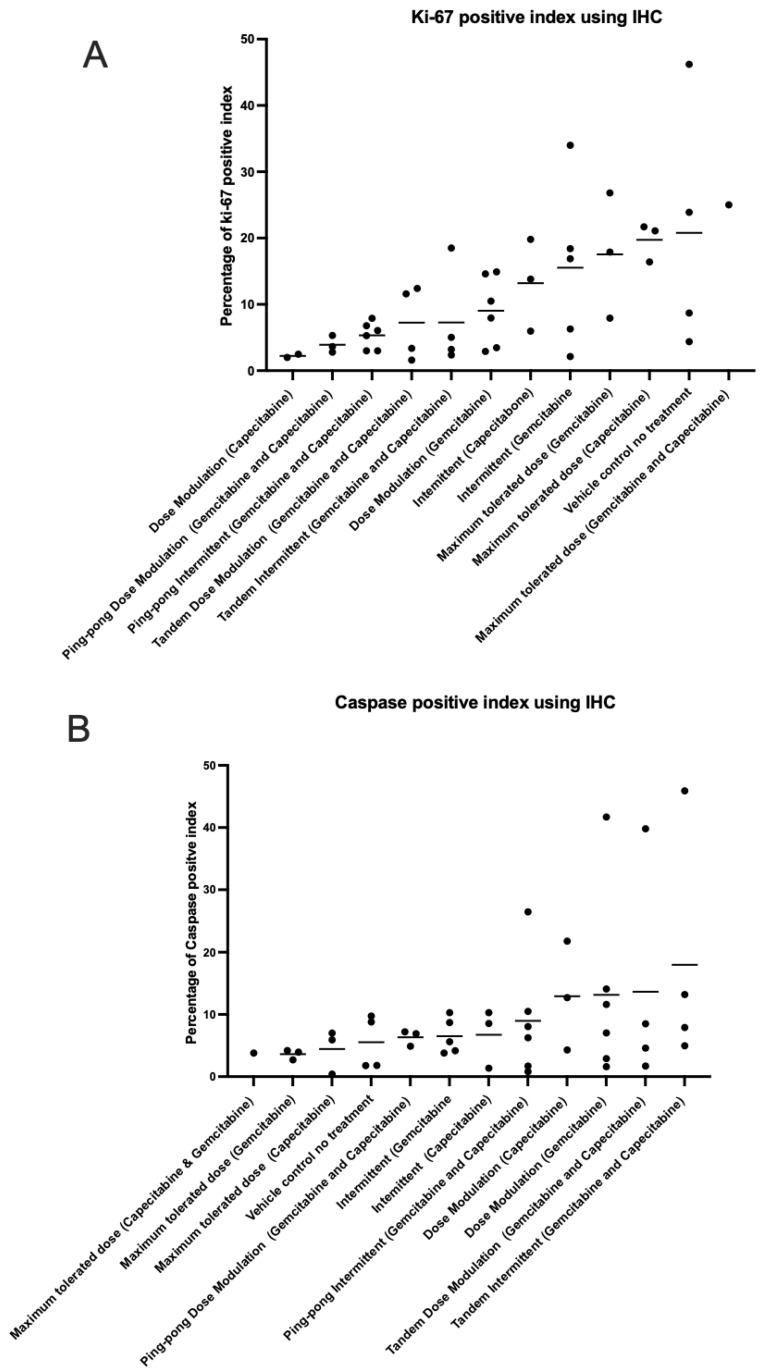
Immunohistochemistry analysis. The percentage of (**A**) proliferating (Ki-67 positive) cells and (**B**) apoptotic (caspase-3 positive) cells in the tumors at the end of the different treatment protocols. Protocols have been ordered by increasing mean values (shown by the bars).

**Figure 9 cancers-16-00257-f009:**
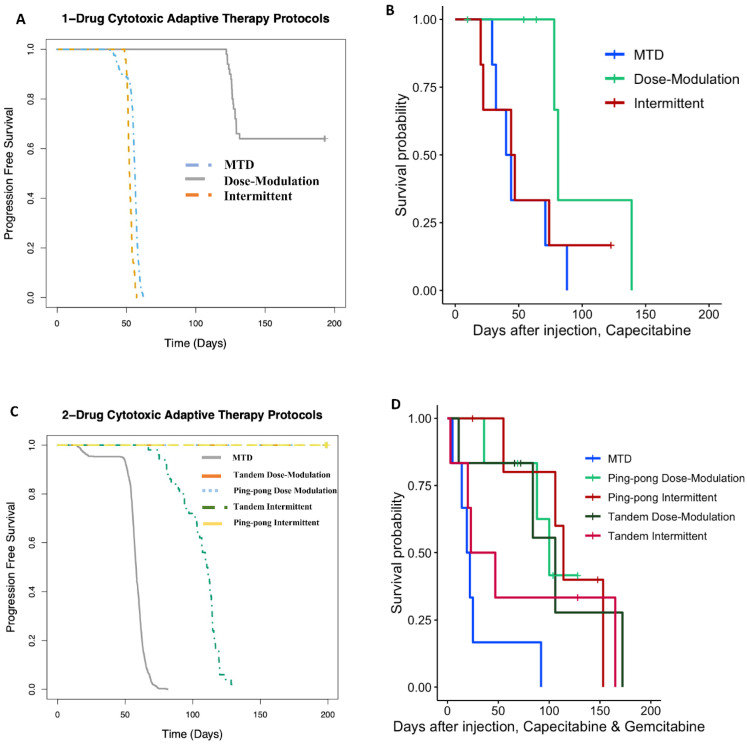
Computational simulations match the rank orders for the different adaptive therapy protocols. Comparison of (**A**) simulation results to (**B**) mouse experimental results for adaptive therapy and MTD protocols using capecitabine alone. (**C**) Comparison of simulation results to (**D**) mouse experimental results for adaptive therapy and MTD protocols using both capecitabine and gemcitabine.

**Table 1 cancers-16-00257-t001:** Treatment and control groups used in this preclinical experiment on endocrine-resistant breast cancer. Each treatment group is shown in different colors: vehicle in light blue, control MTD (single and combination therapy) in gray, gemcitabine single therapy in yellow, capecitabine single therapy in pink, combination ping-pong strategy in dark blue, and combination tandem strategy in dark purple.

Treatment and Control Groups
Control: No treatment control, apply saline (i.p. injection and orally)
0. Control: Standard therapy with maximum tolerated dose (MTD) of gemcitabine
0. Control: Standard therapy with maximum tolerated dose (MTD) of capecitabine
0. Control: Standard therapy with both MTD of gemcitabine and MTD of capecitabine
0. Dose modulation adaptive therapy with gemcitabine
0. Intermittent adaptive therapy with gemcitabine
0. Dose modulation adaptive therapy with capecitabine
0. Intermittent adaptive therapy with capecitabine
0. Ping-pong dose modulation adaptive therapy alternating between gemcitabine and capecitabine
0. Ping-pong intermittent adaptive therapy alternating between both drugs
0. Tandem dose modulation adaptive therapy with both drugs
0. Tandem intermittent adaptive therapy with both drugs

**Table 2 cancers-16-00257-t002:** A. Parameter values for matching the results of the mice experiments using a single cytotoxic drug. B. Parameter values for matching the results of the mice experiments using two cytotoxic drugs.

A	
Parameter	Value
Cell division rate: sensitive	0.06 per hour
Cell division rate: resistant	0.02 per hour
Background death rate	0.01 per hour
Replacement probability	1.0
Delta Tumor	10%
Delta Dose	50%
Probability of death due to drug potency (Ψ)	0.04 per unit drug concentration
Maximum tolerated dose (MTD)	2.5 units
Minimum drug dose	0.5 units
Drug on time	1 h
Frequency of drug application	Once every 24 h
Check tumor burden	Every 3 days
Drug decay	10% per hour
Drug diffusion rate	2.0
Tumor size triggering treatment	Tumor burden is 50% or more of the carrying capacity
Mutation rate	1 × 10^−3^ per cell division
Measurement noise standard deviation (SD)	5 cells
Total grid size	100 by 100
Duration of simulation	5000 h
Stop dosing/initiate treatment vacation when (DM protocols only)	Tumor burden is less than or equal to 25% of carrying capacity
Doubling time of sensitive cells	13.86 h
Doubling time of resistant cells	69.3 h
**B**	
**Parameter**	**Value**
Cell division rate: doubly sensitive	0.10 per hour
Cell division rate: singly resistant	0.06 per hour
Cell division rate: doubly resistant	0.02 per hour
Background death rate	0.01 per hour
Replacement probability	1.0
Delta Tumor	10%
Delta Dose	50%
Probability of death due to drug potency (Ψ)	0.04 per unit drug concentration
Maximum tolerated dose (MTD): Drug 1	2.5 units
Maximum tolerated dose (MTD): Drug 2	2.5 units
Minimum drug dose	0.5 units
Drug on time	1 h
Frequency of drug application	Once every 24 h
Check tumor burden	Every 3 days
Drug decay	10% per hour
Drug diffusion rate	2.0
Tumor size triggering treatment	Tumor burden is 50% or more of the carrying capacity
Mutation rate	1 × 10^−3^ per cell division
Measurement noise standard deviation (SD)	5 cells
Total grid size	100 by 100
Duration of simulation	5000 h
Stop dosing/initiate treatment vacation when (DM protocols only):	Tumor burden is less than or equal to 25% of carrying capacity
Doubling time of doubly sensitive cells	7.7 h
Doubling time of doubly resistant cells	69.3 h
Doubling time of singly resistant cells	13.86 h

## Data Availability

All the data for this study is available in the Appendix A or is freely available from Sareh Seyedi.

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
