# Peer review of "Testing Adaptive Therapy Protocols Using Gemcitabine and Capecitabine in a Preclinical Model of Endocrine-Resistant Breast Cancer"

_cancers, 2024, doi:10.3390/cancers16020257_

Round 1

Reviewer 1 Report

Comments and Suggestions for Authors

This report seeks to explore multiple adaptive chemotherapy dosing regimens in a murine model of breast cancer that is resistant to estrogen and palbociclib. The authors compare their findings with their own mathematical models and find that the outcome correlate.

The question is certainly interesting, from a team who have extensive experience in evolutionary based therapies and cancer evolution more broadly. The experiments are well conducted and their analysis is sound. Unfortunately, their adaptive dosing regimens did not perform terribly well in terms of survival, often actually reducing survival compared to no treatment at all. This does diminish the interest of the article in general, although they do make some observations that they and others will be able to build on in future work. Concerns are listed below

1. The team chose to test dosing schedules using capecitabine and gemcitabine. Although capecitabine is frequently used to treat breast cancer, gemcitabine is a much less commonly used drug. In addition, as far as I am aware, the combination is never used to treat breast cancer. As such it is difficult to understand why they chose this combination as well as their statement: 'We compared gemcitabine to capecitabine to help inform the choice of drug for a future clinical trial.' at lines 120-127. This needs more in depth discussion in the manuscript.

2. They use the same cell line in all their animal experiments. This leaves them open to the criticism that their findings may not be generally applicable and may not be repeatable if they were to evolve resistance again in the same cell line. This limitation should be discussed.

3. Equally, the method they have used to evolve resistance in vitro, while undoubtedly effective, does not really reflect clinical treatment. In clinic the same dose of drug would be used repeatedly, whereas they escalated drug dose, albeit from a high to a very high dose. The rationale for this choice should be explained.

4. Fig 2 shows dose response comparing fulvestrant and palbociclib resistant cells to the parental sensitive cells from which they were derived. It is not clear from the x-axis label or the caption which drug(s) they have applied in this figure.

5. The authors should present dose response curves to gemcitabine and capecitabine in the two cell lines.

6. The authors tell us that ‘20/48 mice treated with one of the adaptive therapy regimens died despite control of tumor size’ (line 394). This is indeed a concern and strongly indicates that the doses/combinations/duration of treatment were toxic. 

In contrast Fig 7 shows that greater drug dose results in more resistant tumors and that bigger tumors at end point are more resistant implying that high dose therapy results in mice dying from large resistant tumors. 

It is important to resolve this conflict in their data. Table S1 shows drug administered, tumor size and cause of death in detail for each mouse. This is a very complete way of showing the data. I would have liked to see a graph (perhaps a bar chart) showing the number of mice in each treatment group that died of each of the causes they list ie: End point tumor size by caliper; Tumor size interfered with walking; Main tumor was small but it showed metastasis to underarm; Died; Skinny/Hunched/Weight loss/weak/unhealthy etc. I realize that numbers will be small but it may be possible to compare toxicity between treatment groups and align that to survival and drug dose administered.

7. The graphs in Fig S1 show bioluminescence as an indicator of total tumour burden. How accurately does luminescence correlate with calliper measurement in their model? Can they please add a supplementary figure?

8. Figure 4 shows that MTD is worse than no treatment. Indeed the authors end by stating, 'The fact that our multidrug “MTD” condition was worse than no treatment suggests that the doses we used for the combined gemcitabine and capecitabine caused a lot of toxicity and were actually above the maximum tolerable dose.' I'm sure they are right but would be more compelling if they could show that this is indeed the case, perhaps as suggested in point 6 above. Can they derive any additional information from their necropsy findings? 

9. The authors used continuous treatment in the MTD regimen. I understand that have done this to mirror the duration of treatment in the adaptive dosing arms. However, this does not accurately reflect how chemotherapy is administered clinically. Capecitabine may be administered for multiple cycles over a long time period but only if the patient is tolerating the drug and responding. Dose reductions would be quite common during long-term therapy. Gemcitabine would only be given for a fixed time period usually 4-6 months. So I think these experiments are actually missing an important control given their aspiration for clinical development. At least, this point should be discussed.

10. I'm not quite sure what I'm looking at in Fig 6. What do the error bars represent? This should be clarified in the caption, together with n and significance. Ideally individual data points should be shown on the graph. Also, why does vehicle treatment induce resistance - can the authors comment?

11. The production in Fig 9 is poor and the font is too small. Please improve.

12. A few typos:

Line 35: 'by' not 'be'

Line 98: 'application' not 'applicaion'

Line 147: 'ping-pong' not 'pin-pong'

Line 527: 'caspase' not 'caspace'

Author Response

Dear Reviewer,

We thank you for your careful reading and helpful suggestions for our manuscript. 

Sincerely,

Sareh Seyedi

Reviewer 2 Report

Comments and Suggestions for Authors

The submitted article entitled: Testing Adaptive Therapy Protocols using Gemcitabine and Capecitabine in a Preclinical Model of Endocrine-Resistant Breast Cancer, presents the very interesting and innovative approach of Adaptive therapy, which could turn cancer into a chronic nonlethal disease, and could change the way that we use current drugs. The methods are clever and robust. Yet there are few points to addressed before further steps:

-Generally, how we can maintain this win-win therapeutic status in patients for longer times?

-What are the limitation of this treatment strategy, can authors discuss the easiness/ difficulty of follow up?

-Methods (2.2.1. Cell Culture):

How cancer cells were authenticated, and from where they were obtained, what is the No. of passages.

-Line 177 what is the percentage of drug-solvent used in the 2nd control? 

Author Response

(The authors gave the same response as above.)
